# Proteomic Analysis of Honey: Peptide Profiling as a Novel Approach for New Zealand Mānuka (*Leptospermum scoparium*) Honey Authentication

**DOI:** 10.3390/foods12101968

**Published:** 2023-05-12

**Authors:** Jessie Bong, Martin Middleditch, Jonathan M. Stephens, Kerry M. Loomes

**Affiliations:** 1School of Biological Sciences and Institute for Innovation in Biotechnology, University of Auckland, Auckland 1010, New Zealand; nbon013@aucklanduni.ac.nz; 2Mass Spectrometry Facility, Faculty of Science, University of Auckland, Auckland 1010, New Zealand; m.middleditch@auckland.ac.nz; 3Comvita NZ Limited, Wilson South Road, Paengaroa, PB1, Te Puke 3119, New Zealand; 4Maurice Wilkins Centre for Molecular Biodiscovery, Auckland 1010, New Zealand

**Keywords:** *Leptospermum scoparium*, mānuka, peptide, parallel reaction monitoring

## Abstract

New Zealand mānuka (*Leptospermum scoparium*) honey is a premium food product. Unfortunately, its high demand has led to “not true to label” marketed mānuka honey. Robust methods are therefore required to determine authenticity. We previously identified three unique nectar-derived proteins in mānuka honey, detected as twelve tryptic peptide markers, and hypothesized these could be used to determine authenticity. We invoked a targeted proteomic approach based on parallel reaction-monitoring (PRM) to selectively monitor relative abundance of these peptides in sixteen mānuka and twenty six non-mānuka honey samples of various floral origin. We included six tryptic peptide markers derived from three bee-derived major royal jelly proteins as potential internal standards. The twelve mānuka-specific tryptic peptide markers were present in all mānuka honeys with minor regional variation. By comparison, they had negligible presence in non-mānuka honeys. Bee-derived peptides were detected in all honeys with similar relative abundance but with sufficient variation precluding their utility as internal standards. Mānuka honeys displayed an inverse relationship between total protein content and the ratio between nectar- to bee-derived peptide abundance. This trend reveals an association between protein content on possible nectar processing time by bees. Overall, these findings demonstrate the first successful application of peptide profiling as an alternative and potentially more robust approach for mānuka honey authentication.

## 1. Introduction

Mānuka honey from New Zealand is a high-value natural food with biomedical applications in wound healing [1]. It is harvested from the indigenous *Leptospermum scoparium* plant and is highly sought after and internationally traded at premium. Mānuka honey is distinguished from non-*Leptospermum* honeys by the presence of the compound, methylglyoxyl, which mediates its unique non-peroxide antibacterial activity [2]. Mānuka honey also harbours an array of other small molecule and phenolic compounds with potential bioactivities [3,4,5,6,7]. Unfortunately, due to its premium market value, mānuka honey is an attractive target for the marketing of non-manuka honeys that are packaged and sold as genuine mānuka honey for increased commercial gain. It is therefore of high priority to develop robust and cost-efficient methods to determine authenticity.

Discovery-based proteomics is a powerful tool for unbiased screening and identification of protein and peptide markers within a sample. Nevertheless, the stochastic nature of data acquisition in order to achieve global analysis also means decreased performance for quantification of the analyte(s) of interest. For a method to be applicable for routine analysis and quality control, it is important that the method can detect the analyte(s) of interest with high precision and sensitivity, yet with minimal interference effects.

To this end, targeted proteomic approaches have been developed for selective monitoring and quantitation of peptide markers. In targeted proteomics, one or more peptides with high selectivity and sensitivity are selected as surrogates, or proxy indicators, of the target protein based on information derived from discovery analysis. The mass analyser is then programmed to focus on the optimal detection and analysis of the candidate peptides based on a predefined *m*/*z* range and retention time window. A number of monitoring methods have been reported for different MS/MS instrument capabilities such as selected reaction monitoring (SRM) on QqQ instruments [8,9,10] and parallel reaction monitoring (PRM), also referred to as high resolution multiple reaction monitoring (MRM-HR), using hybrid Q-TOF or Q-Orbitrap mass spectrometers [11,12,13].

Both SRM and PRM methods demonstrate comparable performance in terms of sensitivity, precision, dynamic range, linearity, and repeatability [14]. Nevertheless, the key to developing a successful SRM method relies heavily on precise determination of the peptide transitions, typically based on empirical evidence or in silico prediction of peptide fragmentation behaviour [8]. In this respect, the PRM method offers an advantage in that it does not require a *priori* selection of target transitions and therefore is relatively easier to develop [14]. Furthermore, PRM-based experiments are highly specific with minimal interference from background ions. In comparison to SRM which typically monitors up to three to five transitions, full MS/MS spectra are acquired during PRM which improves confidence in peptide identification [12]. As one example, the PRM method was successfully applied to authentication of pork protein species in meat mixtures [15].

We have developed several approaches for determining mānuka honey authenticity based on small molecule quantification [5,16,17,18,19,20,21]. We also previously reported a novel proteomic approach that identified unique nectar-derived proteins in New Zealand mānuka (*Leptospermum scoparium*) honey [22]. These proteins were detected as twelve tryptic peptide markers (Table 1) identified based on an in silico-derived mānuka proteome derived from a *Leptospermum* cultivar, *L*. ‘Crimson Glory’ [23]. Eight of these mānuka honey identified peptide markers (PM1–PM8) exhibit 100% sequence identity to tryptic peptides derived from three major mānuka proteins present in the *L*. ‘Crimson Glory’ proteome; a putative pathogenesis-related protein “g10618.t1”, a putative desiccation-related protein “g40782.t1”, and an uncharacterised protein “g7951.t1”. The other four candidate peptide markers (PM1.v1, PM1.v2, PM2 deam., PM7.v1) represent sequence and modified variants of these *L.* ‘Crimson Glory’ peptides.

We hypothesized that these twelve tryptic peptide markers could be used to determine authenticity. In this study, we therefore investigated the use of peptide profiling for New Zealand mānuka honey authentication based on these peptide markers. A PRM-based targeted method was developed for selective monitoring and relative quantification of these peptides (PM1–PM8, PM1.v1, PM1.v2, PM2 deam., and PM7.v1) in honey. Relative abundance of mānuka peptide markers across a range of New Zealand honeys was also investigated. In addition, a selection of six tryptic peptides from three major royal jelly proteins (MRJPs) were analysed to examine their relative abundance across honey types and possible use as internal peptide standards for quantification of mānuka peptide markers.

## 2. Materials and Methods

### 2.1. Honey Samples

Forty-two monofloral and multiforal New Zealand honeys were kindly supplied by Comvita NZ Ltd. comprising mānuka (*L. scoparium*, *n =* 16), kānuka (*Kunzea ericoides*, *n =* 4), rewarewa (*Knightia excelsa*, *n =* 3), kāmahi (*Weinmannia racemosa*, *n =* 4), clover (*Trifolium* spp., *n =* 4), pōhutukawa (*Metrosideros excelsa*, *n =* 3), beech honeydew (*Fucospora solandri*, *n =* 3), ling (*Calluna vulgaris*, *n =* 3), and vipers bugloss (*Echium vulgare*, *n =* 2) honeys. Table A1 shows the composition of all the forty-two honeys with respect to concentrations of approved chemical markers of authenticity: 2’-methoxyacetophenone (2’-MAP), 2-methoxybenzoic acid (2-MB), 3-phenyllactic acid (3-PLA), and 4-hydroxyphenyllactic acid (4-HPA). The supplied mānuka honeys H1-H10 were part of the Unique Mānuka Factor Honey Association (UMFHA) research project to establish monofloral honey standards in collaboration with the honey industry in New Zealand. The 4 chemical markers are present in all the H1–H10 mānuka honeys consistent with a monofloral origin. For all other honeys floral source was assigned by beekeepers based on site analysis and flowering season, and confirmed by assessment of colour, taste, and region of supply by honey identification experts at Comvita. The regional origin, harvest year and protein content of these honeys is shown in Table A2.

### 2.2. Chemicals and Reagents

UltraPureTM Tris was purchased from Invitrogen (Auckland, New Zealand). Promega sequencing grade modified trypsin (PMV5111) was purchased from In Vitro Technologies (Auckland, New Zealand). Thiourea, hydrochloric acid, Tris(2-carboxyethyl)phosphine (TCEP), ammonium bicarbonate, hydrochloride solution trichloroacetic acid (TCA), urea, iodoacetamide, and 1,4-dithiothreitol (DTT) were purchased from Sigma Aldrich (St. Louis, MO, USA). Formic acid was from Scharlau (Barcelona, Spain). The EZQ^®^ Protein Quantitation Kit was from Invitrogen (Auckland, New Zealand). Acetonitrile, acetone and HPLC-grade methanol were from Merck (Darmstadt, Germany). Solid-phase extraction (SPE) cartridges (Strata-X, 33 μm Polymeric Reversed Phase, 10 mg/mL) were purchased from Phenomenex New Zealand (Auckland, New Zealand).

### 2.3. Honey Protein Extraction

TCA chemical precipitation was used to extract honey proteins as described previously [22].

### 2.4. Total Protein Assay

Total protein content of each honey extract was determined using a solid-phase fluorimetric EZQ^®^ assay. A 96-well solid-phase assay plate was assembled as described previously [24]. Re-solubilised honey protein precipitates were diluted 1 in 10 with 5 mM TCEP denaturing buffer (pH 7.0). Chicken ovalbumin was employed as the protein standard.

For each protein standard and sample, an aliquot of 1 μL was spotted onto the protein assay membrane in triplicate. A 5 mM TCEP denaturing buffer was used as a non-protein control. The spotted membrane was left to air-dry for 10 min, followed by washing in 40 mL methanol under gentle agitation for 5 min. The protein membrane was transferred to a staining tray and incubated with 40 mL EZQ^®^ protein quantitation reagent for 30 min under gentle agitation, and washed in rinse buffer (10% methanol, 7% acetic acid) for 2 min. This washing process was repeated two further times and the protein membrane was left to air dry for 90 min.

After drying, the protein membrane was reassembled into the 96-well plate and analysed at ex485–em590 nm using an EnVision^®^ Multilabel Plate Reader (PerkinElmer, Inc., Auckland, New Zealand). Fluorescence intensity was expressed as relative fluorescence units (RFU) with the background signal eliminated by subtracting signal from the non-protein control.

### 2.5. Trypsin Digest

All honey protein extracts were diluted based on measured protein content (Table A2) to a final concentration of 0.25 μg/μL in urea solution (7 M urea and 2 M thiourea in 100 mM Tris-HCl buffer, pH 8). An 80 μL aliquot of the solution, corresponding to 20 μg protein, was used in the trypsin digest and standardised across all samples to allow for quantitative comparison.

Sample reduction was carried out by addition of 2 μL 0.4 M TCEP and incubation in a water bath at 56 °C for 1 h. Following reduction, samples were alkylated with 4 μL 0.4 M iodoacetamide for 30 min in the dark at room temperature. 1 M DTT (2 μL) was used to quench the iodoacetamide reaction.

A working trypsin solution was prepared from a stock solution (1 μg/μL trypsin) prepared in resuspension buffer and diluted ten-fold with 50 mM ammonium bicarbonate. Prior to trypsin digest, the protein solution was diluted with 50 mM ammonium bicarbonate to a final volume 500 μL. This dilution step decreases the amount of urea present in the protein solution to <1 M for trypsin digest. To each sample 1 μg of trypsin was added (1:20 trypsin:protein ratio) followed by incubation for 12 h at 37 °C. 10% aqueous formic acid (50 μL) was used to terminate Trypsin activity.

### 2.6. Solid-Phase Extraction

Following trypsin digestion, a solid-phase extraction (SPE) step was performed as described previously [22].

### 2.7. PRM Analysis

A targeted PRM method was developed on a Triple-TOF 6600 nanoLC-QqTOF-MS/MS system (AB Sciex, Victoria, Australia) at the Mass Spectrometry Facility, Faculty of Science, Auckland, University of Auckland.

Prior to analysis, each reconstituted peptide concentrate was diluted 20-fold with 0.1% aqueous formic acid. An injection volume of 3 μL was applied. Separation was performed on a ReproSil-Pur 120 trap column (10 × 0.3 mm; ESI Source Solutions, Woburn, MA, USA) coupled to an in-house packed ReproSil-Pur 120 picofrit column (3 μm; 150 mm × 75 μm; ESI Source Solutions, Woburn, MA, USA). The binary mobile phase consisted of 0.1% aqueous formic acid (Solvent A) and acidified acetonitrile containing 0.1% formic acid (Solvent B). The flow rate was 300 nL/min. A 45-min gradient elution programme was applied: initial (5% B), 32 min (40% B), 34 min (95% B, held 3 min), 38 min (5% B, held 7 min).

Peptides were detected in PRM mode at an ion spray voltage of 2.7 kV, curtain gas of 35 unit, nebuliser gas of 25 unit, interface heater temperature of 150 °C, focusing voltage of 10 V, and declustering potential of 80 V. Twelve mānuka peptide markers, inclusive of the sequence variants and deamidated form, were included in the PRM method, together with some bee-derived MRJP tryptic peptides (Table 1). Acquired mass spectral data were processed using Skyline software (Version 4.1) (MacCoss Lab Software, Seattle, WA, USA).

### 2.8. Spectral Library Data Acquisition

To acquire a representative mānuka elution profile, an aliquot of reconstituted peptide concentrate from each of the 10 mānuka honey protein samples, H1–H10 (Table A1 and Table A2), were combined. This bulk sample was analysed using a non-targeted full scan using information-dependent acquisition (IDA) mode on the same Triple-TOF 6600 nanoLC-QqTOF-MS/MS system (AB Sciex, Victoria, Australia). The acquired MS/MS data from this bulk mānuka honey sample was used as a reference to confirm target peptide identity using PRM analysis and served as the spectral library for processing the PRM data using Skyline software (Version 4.1) (MacCoss Lab Software, Seattle, WA, USA).

Prior to analysis, the representative mānuka honey protein sample (H1–H10) was diluted 20-fold with 0.1% aqueous formic acid. An injection volume of 5 μL was applied. Chromatographic separation was performed as described above (Section 2.7).

Mass spectra were acquired as described previously [22].

### 2.9. Statistical Analysis

Statistical analyses were performed using Graphpad Prism software (Version 7.01) (GraphPad Software Inc., San Diego, CA, USA). Correlations were determined by regression analysis. Comparison between multiple group means were performed by one-way ANOVA. Where the overall significance (*p*-value) is less than 0.05, a Tukey’s multiple comparison post-hoc test was carried out to compare the difference between individual groups.

## 3. Results & Discussion

### 3.1. Relative Abundance of Peptide Markers in Mānuka Honey

All mānuka peptide markers, PM1–PM8, and corresponding 4 variants (PM1.v1, PM1.v2, PM2 deam., PM7.v1) were detected in the 16 mānuka honeys (H1-H10 & H37-H42) examined. Peptide identities were verified based on retention time, peak intensity ranking and the relative ratios of the fragment ions using the Skyline software interface (v2.7 and v2.8). Figure 1A–C show representative extracted ion chromatograms (XICs) of the targeted peptide precursor ions (top panel) and corresponding selection of fragment ions for quantification (bottom panel). The degree of match between the observed peptide transition spectra and MS/MS spectral library is indicated by dot product (dotp) values. A high dotp value indicates a good peptide-spectral match and absence of interfering signal, with the maximum dotp value being 1.0.

Each peptide precursor ion exhibits a set of fragment ions with specific peak intensity ranking as indicated by the colour-coded fragments (bottom panel, Figure 1A–C). This analysis allows confirmation of peptide identity while the peak area sum of the fragment ion selection for each peptide corresponds to the peptide’s relative abundance.

As the targeted peptides are fragmented using selected collision energies under identical conditions, it is possible to determine the relative abundance of these peptides based on their respective signal intensity. For each mānuka honey sample, a standardized 20 μg aliquot of protein from the honey precipitate was subjected to tryptic digest. This standardization procedure ensured consistent protein loading for analysis across samples thereby eliminating inter-sample protein content variability.

Figure 2 shows relative abundance of the twelve mānuka peptide markers in the 16 mānuka honeys examined based on total integrated peak area of the selected fragment ions. Relative abundance of the mānuka peptide markers varied across samples, with most peptides exhibiting three- to seven-fold difference. Nevertheless, at the protein level (g7951.t1, g10618.t1, g40782.t1), the recorded variance was relatively constant with peptides derived from the same protein exhibiting similar fold variance. For example, both peptide markers from the putative pathogenesis-related protein g10618.t1, PM4 (ANYPPYGIDFPAGATGR, *m*/*z* 883.93, z +2) and PM5 (ISLNSQLQNHLVTISR, *m*/*z* 608.34, z +3), exhibited approximately five-fold variance, whereas PM6 (GSIGQGLDSIAPYLAQGGPQPVG-AAK, *m*/*z* 818.10, z +3), PM7 (IVYGSGSEYKPGGFYPDGGNGR, *m*/*z* 759.69, z +3), and PM8 (TTSNVLSADVDSVSYSR, *m*/*z* 600.96, z +3) derived from the putative desiccation-related protein. g40782.t1, and the sequence variant PM7.v1 (IVYGSGSEYKPGGFYPAGGNGR, *m*/*z* 745.03, z +3) exhibited three- to four-fold variance.

For the uncharacterised mānuka protein, g7951.t1, the peptide marker PM1 (CLLLFFPGLNTR, *m*/*z* 725.90, z +2) and PM3 (VINGLSTKCLLLFFPGLNTR, *m*/*z* 755.09, z +3) exhibited approximately seven-fold variance. The sequence variants PM1.v1 (ALLLFFPGLNTR, *m*/*z* 681.40, z +2) and PM1.v2 (SLLLFFPGLNTR, *m*/*z* 689.40, z +2) also exhibited similar variance in the range of five- to six-fold. However, PM2 (VINGLSTK, *m*/*z* 416.25, z +2) and its deamidated form, PM2 deam.(VINGLSTK deamidated, *m*/*z* 416.74, z +2) exhibited almost twice the variance at eleven- and fourteen-fold, respectively. This higher variance recorded for PM2 and its deamidated counterpart may be due to either random artefactual deamidation or biologically-regulated deamidation of one form to the other.

There was variance in peptide abundance among mānuka honeys harvested from Northland, Waikato, East Coast, Whanganui, and Wairarapa (Table A2). Five out of the six peptide markers representing protein g7951.t1, including the sequence and deamidated variants, were detected at increased abundance in honeys harvested from the Whanganui region (*p* < 0.01) except for PM3 (VINGLSTKCLLLFFPGLNTR, *m*/*z* 755.09, z +3) where no difference was recorded across regions (*p* > 0.05) (Figure A1A–F). The protein g10618.t1-derived peptide marker, PM4 (ANYPPYGIDFPAGATGR, *m*/*z* 883.93, z +2), was present at significantly greater abundance in Northland and Waikato compared to East Coast and Wairarapa samples (*p* < 0.05) (Figure A1G), whereas PM5 (ISLNSQLQNHLVTISR, *m*/*z* 608.34, z +3) was present at significantly greater abundance in the Northland, Waikato, and Whanganui samples (*p* < 0.05) (Figure A1H). For peptide markers from protein g40782.t1 (Figure A1I–L), the Whanganui samples also carried elevated abundance for most peptides except for PM8 (TTSNVLSADVDSVSYSR, *m*/*z* 600.96, z +3) which occurred at higher relative abundance in the Northland and Wairarapa samples.

This variance in peptide abundance is likely due to factors other than intrinsic regional influence. Taking into account the small sample size, there is wide variability in relative peptide marker abundance across regions. This variance likely reflects differing proportions of mānuka nectar within the honeys due to availability of other floral sources. In his regard, the nectar sources harvested by honey bees will be heavily influenced by both apiary placement and local proximity of *L. scoparium* trees.

### 3.2. Comparison to Other New Zealand Honeys

Analysis of other New Zealand honey varieties showed negligible traces of the mānuka tryptic peptide markers, PM1–PM8 and variants. For visualization of these peptide abundance data, a heat map was constructed based on normalized peak area (Figure 3). For each peptide, the peak area exhibited by each sample was adjusted by normalisation to the greatest peak area recorded for that peptide, and relative peptide abundance indicated by a colour intensity scale reflecting the percentage of greatest peak area it constitutes.

The graphical illustration of relative peptide abundance in Figure 3 clearly distinguished mānuka (*L. scoparium*) from non-mānuka honeys based on the twelve selected mānuka peptide markers. Mānuka honey evidently carried higher abundance of all peptide markers in comparison to the non-mānuka honeys, which mostly displayed negligible traces of the mānuka peptides.

Peptide markers derived from the uncharacterised mānuka protein g7951.t1, namely PM1–PM3 and the corresponding peptide variants, were virtually absent in all non-mānuka honeys except for one kānuka (*K. ericoides*) (sample H13), one pōhutukawa *(M. excelsa*) (sample H26), and one beech honeydew *(F. solandri*) (sample H31) honey. These three honeys exhibited low levels of the mānuka peptides at less than 5% relative abundance and likely reflects a minor mānuka nectar contribution due to floral dilution.

Similarly, the two peptide markers PM4 and PM5 derived from the putative pathogenesis-like protein g10618.t1 were not detected collectively in clover (*Trifolium* spp.), ling (*C. vulgaris*), and vipers bugloss (*E. vulgare*) honeys. Rewarewa (*K. excelsa*), kāmahi (*W. racemosa*). Two-thirds of the pōhutukawa (*M. excelsa*) and beech honeydew *(F. solandri*) honeys exhibited up to 5% relative abundance of these peptides while pōhutukawa, beech honeydew, and three quarters of the kānuka (*K. ericoides*) honeys exhibited up to 9%.

Peptide markers PM6–PM8 and PM7.v1 representing the putative desiccation-related protein g40782.t1 were also absent in clover (*Trifolium* spp.), ling (*C. vulgaris*), and vipers bugloss (*E. vulgare*) honeys. By comparison, these peptides were detected in two of the three pōhutukawa (*M. excelsa*) honeys (4–7% relative abundance), in beech honeydew (*F. solandri*) honey sample H31 (12% relative abundance for PM8), and in kānuka (*K. ericoides*) honeys (7–26% relative abundance with exception of one sample (H14)).

On the basis of these comparisons of mānuka and non-mānuka honeys, it appears that peptide markers derived from protein g7951.t1 (PM1–PM3, PM1.v1, PM1.v2, PM2 deam.) are more definitive for mānuka honey with minimal occurrence in the other honey types. Nevertheless, the overall relative abundance of these peptides within the mānuka honey group was comparatively lower than those of peptide markers from protein g10618.t1 and g40782.t1, with the majority of samples exhibiting relative abundance in the range of 20–50% of the highest respective abundant sample. Peptide markers from protein g10618.t1 and g40782.t1 occurred predominantly within this mānuka honey sample set at more than 40% relative abundance.

The PRM approach reported herein provides a method for routine monitoring of mānuka peptide markers in honey. Analysis of a range of monofloral New Zealand honey varieties demonstrated negligible traces of mānuka peptides in non-mānuka honeys. Low levels of these peptides may be present in some non-mānuka honeys due to floral dilution, especially in particular honey types harvested from areas that contain *L. scoparium* such as kānuka honey, and to a lesser extent, pōhutukawa honey.

Whilst all honeys in this study were stored at 4 °C following their collection, the influence of honey age on total peptide abundance was examined based on the harvest year. Here, a general trend was observed whereby fresher honeys (1 year old) exhibited on average a greater sum of relative peptide abundance compared to older honeys that have been stored in the chiller for 10 and 11 years (Figure A2). This may reflect seasonal variation, or possibly, protein degradation during storage as supported by a shift in protein profile for comparatively older honeys [25]. Chemical modifications involving proteins have been reported to occur in aging honeys, such as interaction with phenolics and polyphenolics [7,26], or formation of Maillard reaction products [27]. The lack of interim aged mānuka honeys prevented further analysis of this effect in this study.

Overall, the suite of identified mānuka peptide markers (PM1–PM8 and variants) were reproducibly detectable and quantifiable in mānuka honey, the floral purity of which could be estimated with reference to a sufficiently comprehensive database. Incorporation of these peptides as potential markers for mānuka honey authentication should account for potential seasonal variation as well as peptide marker stability over time.

### 3.3. MRJP Peptide Relative Abundance across Honey Types

Whilst the mānuka tryptic peptide markers were mostly confined to mānuka honey, the MRJP tryptic peptides were detected in all the New Zealand honeys analysed (Figure A3). There was some variability in MRJP tryptic peptide relative abundance, but the ratios within a protein were relatively consistent across samples (Figure A3). Ling (*C. vulgaris*) honey displayed lower MRJP tryptic peptide abundance compared to the other honeys, nevertheless these differences were considered non-diagnostic. Overall, no consistent trend was discernible between MRJP tryptic peptide relative abundance and floral source for the honey types analysed.

Analysis of the sum of targeted nectar- and bee-derived tryptic peptide relative abundance showed a varying proportion of mānuka and bee peptides across the mānuka honey samples (Figure 4A). The sum of mānuka peptides was calculated as the total relative abundance of PM1–PM8 and the peptide variants PM1.v1, PM1.v2, PM2 deam., and PM7.v1, whereas the sum of bee peptides comprised total relative abundance of the MRJP tryptic peptides BP1–BP6.

Mānuka honeys with a lower total protein content appear to carry a higher relative proportion of nectar- to bee-derived peptide abundance, and vice versa (Figure 4B). For example, the sum of mānuka peptides was approximately two-fold greater than that of the bee peptides in mānuka protein extracts, H41 and H42, which carried <0.5 μg/μL protein content. The 0.5–1.0 μg/μL mānuka protein extract group exhibited a relative mānuka:bee peptide proportion of one- to two-fold, whereas the >1.0 μg/μL group mostly exhibited a greater proportion of bee-derived peptides. Regression analysis of the mānuka:MRJP peptide sum ratio and the protein content of the mānuka honey precipitate yielded a non-linear regression that was best-fitted to an exponential decrease function (Figure 4B).

This observed trend between total protein content and relative abundance of the targeted mānuka and MRJP peptides may reflect the extent to which bee-derived proteins are incorporated into the ripening nectar by the honey bees. Possibly, honey bees spend extended processing time on nectars with a higher protein content in order to mature the right consistency prior to deposition into the honeycomb cells. The increased exposure time of nectar to the bee hypopharyngeal gland secretions may then result in a higher amount of bee proteins incorporated into the nectar. This possibility supports a previous suggestion that proline derived from the hypopharyngeal gland may aid in maturing consistency by counteracting the osmotic pressure of nectar [28].

The observed variation in MRJP peptide content across honeys precludes their use as internal standards for quantification of mānuka peptide markers. Nevertheless, whilst bee-derived peptides are considered non-diagnostic for New Zealand honey floral sources, their reproducible presence in honey can be employed as a metric to identify fraudulently labelled sugar syrup, which lacks these peptides, or possibly for detection of substantial post-harvest sugar syrup adulteration through markedly reduced MRJP peptide abundance. For the latter application, it may require analysis of a larger sample set to establish a working range of MRJP tryptic peptide abundance in honey, especially if the content of these peptides incorporated in honey by honey bees is influenced by nectar sugar and protein content.

## 4. Conclusions

This study reports a highly sensitive PRM mas spectrometry method that could be used for routine monitoring and relative quantification of mānuka peptide tryptic markers in honey. All twelve mānuka peptide tryptic markers (PM1–PM8, PM1.v1, PM1.v2, PM2 deam., and PM7.v1) were present in mānuka honeys, but primarily absent in the other honey types. Floral dilution of *L. scoparium* nectar likely explains the observed traces of these mānuka peptides in non-mānuka honeys as well as their regional variance. By comparison, the six bee-derived MRJP tryptic peptides were consistently detected across all New Zealand honey types but with sufficient variation in relative abundance that precluded their utility as internal standards. Interestingly, the ratio of summed mānuka nectar:MRJP peptides intensities was inversely correlated to total protein content in the honey protein extracts. This observation reveals an association between endogenous protein content of *L. scoparium* nectar and the absolute bee protein content that is ultimately incorporated by the honey bees during the nectar-to-honey conversion process. This study presents the first successful application of peptide profiling to characterise honey floral origin. The described PRM method could be potentially adopted for routine monitoring of mānuka peptide tryptic markers in honey in testing laboratories as an alternative proteomic approach to New Zealand mānuka (*L. scoparium*) honey authentication.

## Figures and Tables

**Figure 1 foods-12-01968-f001:**
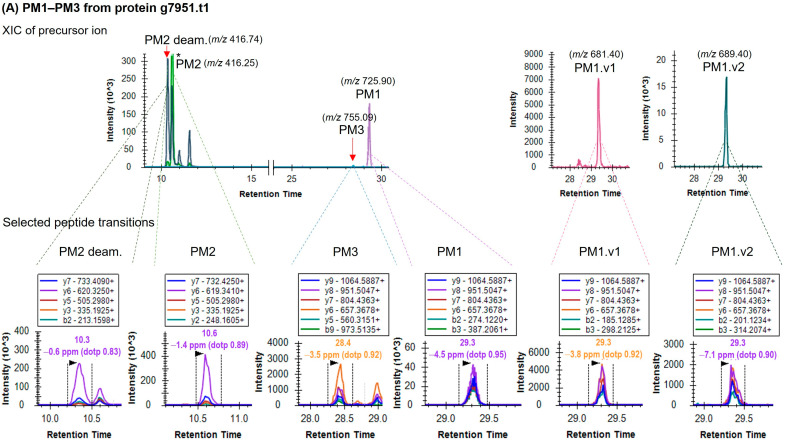
Extracted ion chromatograms (XICs) of mānuka peptide transitions for (**A**) protein g7951.t1, (**B**) protein g101618.t1, and (**C**) protein g40782.t1 in mānuka honey H1. Top panels show precursor ion XIC and bottom panels show corresponding fragment ion selection. Dotted lines on either side of fragment ion peaks indicate integration boundaries. Retention time (min), mass error (ppm), and dotp value are displayed above fragment ion peaks. Asterisk (*) indicates a truncated peak.

**Figure 2 foods-12-01968-f002:**
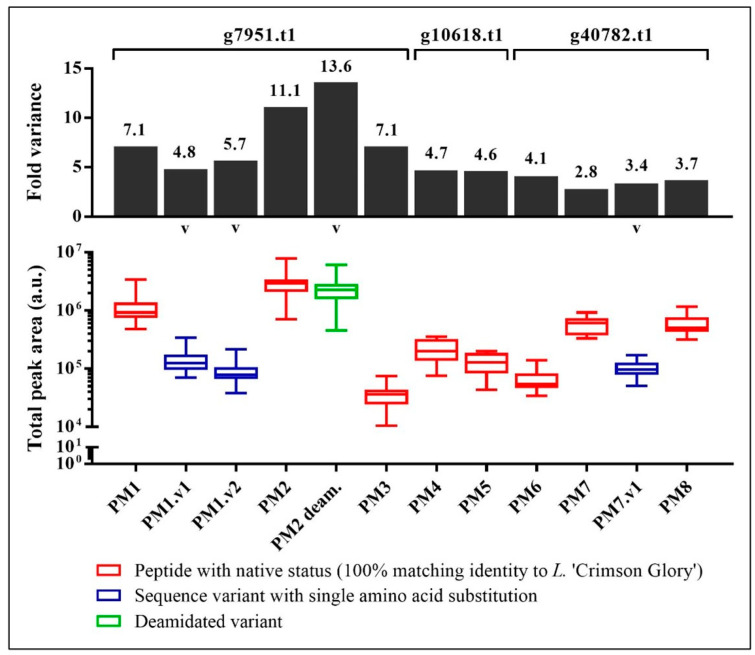
Relative abundance of targeted tryptic peptide markers in mānuka honey (*n =* 16) as measured by total peak area and corresponding fold variance observed for each peptide. Variants denoted by ‘v’.

**Figure 3 foods-12-01968-f003:**
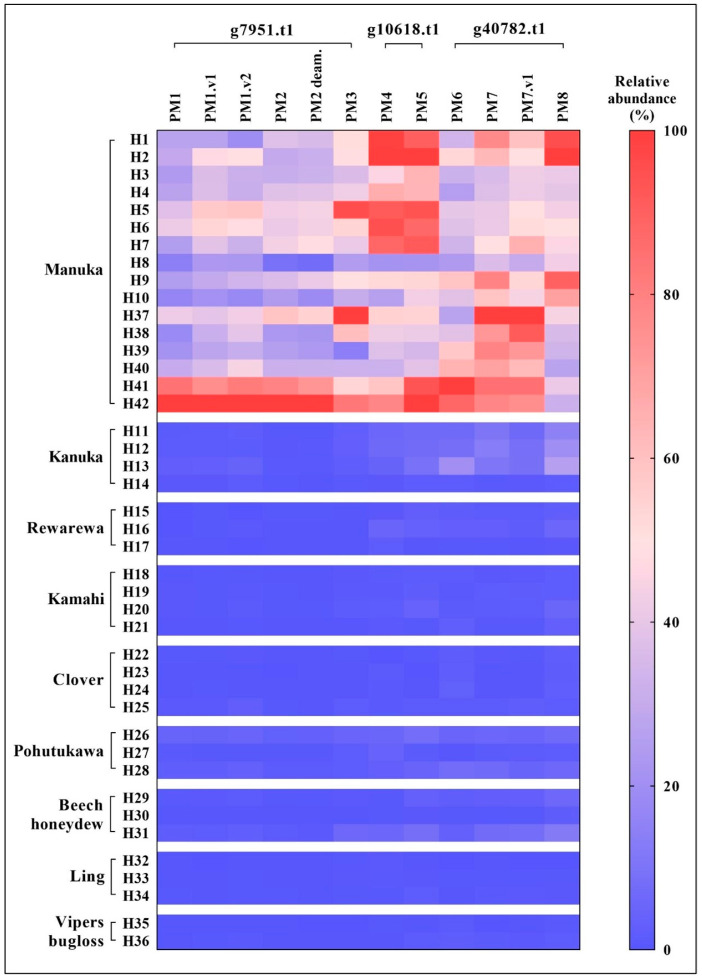
Heat map showing relative abundance of mānuka tryptic peptide markers across nine New Zealand honey types based on normalized total peak area. Relative abundance expressed as percentage of greatest peak area for each peptide marker.

**Figure 4 foods-12-01968-f004:**
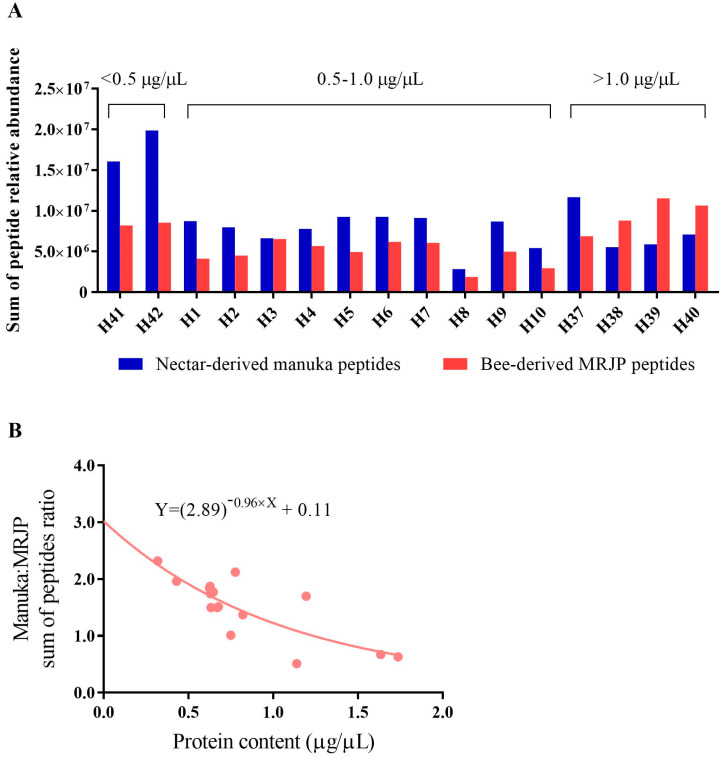
(**A**) Sum of targeted nectar- and bee-derived peptide intensities in 16 mānuka honeys. For these analyses a standardized 20 μg protein sample from each honey was subjected to tryptic digest. Mānuka peptides encompassed PM1–PM8 and the peptide variants PM1.v1, PM1.v2, PM2 deam., and PM7.v1. Bee peptides represented by BP1–BP6. The original raw honey total protein content (μg/μL) is indicated above the bars. (**B**) Non-linear regression analysis between mānuka:MRJP peptide sum ratio and protein content across honey extracts.

**Table 1 foods-12-01968-t001:** Unique nectar- and bee-derived MRJP tryptic peptides identified in New Zealand mānuka honey. ^a^ Red font indicates modifications to predicted *L.* ‘Crimson Glory’ peptide sequence, bold font indicates missed tryptic cleavage which is not associated with an adjacent proline residue.

No.	Targeted Peptide	Sequence ^a^	Protein	Precursor *m/z*	*z*	Collision Energy (kV)	Accumulation Time (ms)
1	PM1	CLLLFFPGLNTR	g7951.t1	725.90	2	42	20.0
2	PM1.v1	ALLLFFPGLNTR	g7951.t1	681.40	2	40	15.0
3	PM1.v2	SLLLFFPGLNTR	g7951.t1	689.40	2	43	20.0
4	PM2	VINGLSTK	g7951.t1	416.25	2	23	10.0
5	PM2 deam.	VINGLSTK deamidated	g7951.t1	416.74	2	23	10.0
9	PM3	VINGLSTKCLLLFFPGLNTR	g7951.t1	755.09	3	42	100.0
7	PM4	ANYPPYGIDFPAGATGR	g10618.t1	883.93	2	52	30.0
8	PM5	ISLNSQLQNHLVTISR	g10618.t1	608.34	3	34	25.0
9	PM6	GSIGQGLDSIAPYLAQGGPQPVGAAK	g40782.t1	818.10	3	46	30.0
10	PM7	IVYGSGSEYKPGGFYPDGGNGR	g40782.t1	759.69	3	42	20.0
11	PM7.v1	IVYGSGSEYKPGGFYPAGGNGR	g40782.t1	745.03	3	42	30.0
12	PM8	TTSNVLSADVDSVSYSR	g40782.t1	600.96	3	33	20.0
13	BP1	SLPILHEWK	MRJP1	561.82	2	32	10.0
14	BP2	LLTFDLTTSQLLK	MRJP1	746.93	2	44	10.0
16	BP3	HIDFDFGSDER	MRJP3	669.29	2	39	20.0
17	BP4	GGPLLRPYPDWSFAK	MRJP3	568.64	3	31	20.0
18	BP5	IINNDFNFNDVNFR	MRJP3	871.42	2	51	20.0
19	BP6	YLDYDFGSDER	MRJP5	690.29	2	40	20.0

## Data Availability

The datasets generated for this study are available on request to the corresponding author.

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
