# Peer review of "Proteomic Analysis of Honey: Peptide Profiling as a Novel Approach for New Zealand Mānuka (Leptospermum scoparium) Honey Authentication"

_foods, 2023, doi:10.3390/foods12101968_

Round 1

Reviewer 1 Report

Dear Authors,

After reading your manuscript, I realized that you examined the Proteomic analysis of honey as peptide profiling as a novel approach for NZ mānuka (Leptospermum scoparium) honey authentication. The issue that you dealt with is well described in the paper, and your work complements previous knowledge and opens up the possibility for further research. The paper is technically and clearly written. The discussion is clearly written. The methodological details are explained well. Also, the figures are very clearly presented.

Remarks: The linguistic quality of the paper should be revised more carefully once more by an English expert.

The title summarizes the organization of the work well. On the contrary, the abstract deserves further study, especially for the part relating to the results.

The Introduction does not indicate the status of current knowledge. Moreover, there doesn’t seem to be a clear research hypothesis formulated. 

The experimental design is appropriate to resolve the stated objectives of the study. The experimental techniques are appropriate to resolve the stated objectives of the study.

Certainly, the results are important to scientific literature.

Sincerely,

Reviewer 2 Report

The manuscript deals with the analysis of honey peptides as a tool for honey authentication and manuka honey identification. The paper provides novel information and is of practical importance. However, I have some comments listed below:

L2: New Zealand instead NZ

Abstract:

Add some values of examined parameters or % changes

Introduction:

Add a first paragraph related to the importance and properties of honey and manuka honey. Include composition of vitamins, phenolic acids, flavonoids, proteins, carbohydrates, amino acids, royal jelly aliphatic acids. For this purpose the Authors may refer to the following references: https://doi.org/10.2478/jas-2018-0012, https://doi.org/10.3390/ijerph20032458.

Materials and methods:

L184-190: the mobile phase and MS/MS conditions were the same as previously described (p. 2.7), therefore, the description in p. 2.8 can be omitted

Results and discussion:

This part is actually a description of the results and lacks a discussion based on other references.

Fig. A1 and A2 should be renamed to Tables A1 and A2. Means of protein and acid contents should be added for each honey type.

Why the results from Fig. A1 and A2 are not described in the text?

‘Relative abundance’ and ‘peak area’ in Figures should be replaced by concentration

Conclusions:

Shorten the Conclusions and summarize only the main results.

L426: HPLC-UV, but this method is missing in Materials and methods section

Reviewer 3 Report

Important study about verification of manuka honey. Manuscript is well written. I did only several comments in attached PDF file.

My main concern is about the verification of monofloral origin of the tested honey samples when no microscopy or any other analysis were done! Authors has to be able to prove that samples were really monofloral.

The authors confused correlation with regression analysis. Yes, result is really interesting for honey in general, however, "r" is not "R2" and equation shuld be given. Centrifugation has to by specified with "g" rpm is not sufficient.

Reviewer 4 Report

In this work, a highly sensitive parallel reaction monitoring method was developed for routine monitoring and relative quantification of mānuka peptide markers in honey. The use of peptide profiling for New Zealand mānuka honey authentication based on these twelve identified mānuka peptide markers was investigated. The method is interesting and novel, and it needs moderate revision. Herein, some detailed comments are addressed as follows:

1. The keywords, Parallel reaction monitoring and PRM were repetitive.

2. Line 52, in this study? Check the sentence.

3. Line 81, Figure A1?, it should be a table. And the Figure A2 is the same.

4. Line 102, -80 ºC should be -80 ℃.

5. Line 113 and 115, “gentle agitation”, please describe it.

6. Figure 1 is illegible.

7. The tables and figures in Appendix A were difficult to distinguish. Please revise the names to Table S? or Figure S?.

8. The line 278, the color of the “tryptic” is red.

9. Line 324, the fresher honeys exhibited on a greater sum of relative peptide abundance compared to older honeys, however, the relative peptide abundance in fresher honeys H37-H40 is relatively close to the most of the older honeys. Why? And how to enhance the accuracy of ? 

10. Why the PRM method was chosen? Please explain it in detail.

11. The Nectar-derived peptides and Bee-derived peptides were lack of clearer descriptions.

12. Paragraph 1 in Page 11, “assuming…” The explanation about protein content trend is not clear, please revise it.

13. Please discuss more possibility usage for this method in the conclusion section. 
